# UBL3 Interacts with Alpha-Synuclein in Cells and the Interaction Is Downregulated by the EGFR Pathway Inhibitor Osimertinib

**DOI:** 10.3390/biomedicines11061685

**Published:** 2023-06-10

**Authors:** Bin Chen, Md. Mahmudul Hasan, Hengsen Zhang, Qing Zhai, A. S. M. Waliullah, Yashuang Ping, Chi Zhang, Soho Oyama, Mst. Afsana Mimi, Yuna Tomochika, Yu Nagashima, Tomohiko Nakamura, Tomoaki Kahyo, Kenji Ogawa, Daita Kaneda, Minoru Yoshida, Mitsutoshi Setou

**Affiliations:** 1Department of Cellular and Molecular Anatomy, Hamamatsu University School of Medicine, 1-20-1 Handayama, Higashi-Ku, Hamamatsu 431-3192, Shizuoka, Japan; chenbin19101@gmail.com (B.C.); hasan.mahmudbio@gmail.com (M.M.H.); d19110@hama-med.ac.jp (H.Z.); zhaiqing199404@gmail.com (Q.Z.); wali.rubmb10@gmail.com (A.S.M.W.); pingyashuang1989@gmail.com (Y.P.); zhangchi07.pegasus@gmail.com (C.Z.); oyamasoho@gmail.com (S.O.); afsana.mimichem@gmail.com (M.A.M.); hhelibeyuna@gmail.com (Y.T.); kahyo@hama-med.ac.jp (T.K.); 2Institute for Medical Photonics Research, Preeminent Medical Photonics Education and Research Center, Hamamatsu University School of Medicine, Hamamatsu 431-3192, Shizuoka, Japan; yunaga@hama-med.ac.jp; 3Department of Neurology, Hamamatsu University School of Medicine, 1-20-1 Handayama, Higashi-Ku, Hamamatsu 431-3192, Shizuoka, Japan; tomohiko@hama-med.ac.jp; 4International Mass Imaging Center, Hamamatsu University School of Medicine, 1-20-1 Handayama, Higashi-Ku, Hamamatsu 431-3192, Shizuoka, Japan; 5Laboratory of Veterinary Epizootiology, Department of Veterinary Medicine, Nihon University, Kameino 1866, Fujisawa 252-0880, Kanagawa, Japan; ogawa.kenji@nihon-u.ac.jp; 6Choju Medical Institute, Fukushimura Hospital, Yamanaka-19-14 Noyoricho, Toyohashi 441-8124, Aichi, Japan; kaneda@chojuken.net; 7Department of Biotechnology, Graduate School of Agricultural and Life Sciences, The University of Tokyo, Tokyo 113-8657, Japan; yoshidam@riken.jp; 8Collaborative Research Institute for Innovative Microbiology, The University of Tokyo, Tokyo 113-8657, Japan; 9RIKEN Center for Sustainable Resource Science, Wako 351-0198, Saitama, Japan; 10Department of Systems Molecular Anatomy, Institute for Medical Photonics Research, Preeminent Medical Photonics Education & Research Center, 1-20-1 Handayama, Higashi-Ku, Hamamatsu 431-3192, Shizuoka, Japan

**Keywords:** UBL3, α-synuclein, interaction, drug screen, EGFR pathway inhibitor, osimertinib, downregulate, α-synucleinopathies

## Abstract

Ubiquitin-like 3 (UBL3) acts as a post-translational modification (PTM) factor and regulates protein sorting into small extracellular vesicles (sEVs). sEVs have been reported as vectors for the pathology propagation of neurodegenerative diseases, such as α-synucleinopathies. Alpha-synuclein (α-syn) has been widely studied for its involvement in α-synucleinopathies. However, it is still unknown whether UBL3 interacts with α-syn, and is influenced by drugs or compounds. In this study, we investigated the interaction between UBL3 and α-syn, and any ensuing possible functional and pathological implications. We found that UBL3 can interact with α-syn by the *Gaussia princeps* based split luciferase complementation assay in cells and immunoprecipitation, while cysteine residues at its C-terminal, which are considered important as PTM factors for UBL3, were not essential for the interaction. The interaction was upregulated by 1-methyl-4-phenylpyridinium exposure. In drug screen results, the interaction was significantly downregulated by the treatment of osimertinib. These results suggest that UBL3 interacts with α-syn in cells and is significantly downregulated by epidermal growth factor receptor (EGFR) pathway inhibitor osimertinib. Therefore, the UBL3 pathway may be a new therapeutic target for α-synucleinopathies in the future.

## 1. Introduction

Ubiquitin-like 3 (UBL3), a highly conserved ubiquitin-like protein first found in eukaryotes, is localized to the cell membrane by prenylation [1]. The UBL3 gene is widely expressed in human tissues, with the strongest expression in the testis, ovary, and brain tissues [2]. In our previous study, UBL3 was characterized as a post-translational modification (PTM) factor that regulates protein sorting to small extracellular vesicles (sEVs) [3]. Another recent study found that UBL3 is involved in adaptive immunity by regulating the transport of major histocompatibility complex II and CD86 through ubiquitination [4]. The downregulated expression of UBL3 has been reported to be associated with human diseases, such as cervical cancer [5], gastric cancer [6], esophageal cancer [7], and non-small-cell lung cancer [8]. Moreover, UBL3 can interact with more than 22 disease-related proteins, including neurodegenerative-disease-related molecules [3]. The previous proteomic analyses, however, may be insufficient since the results were only from MDA-MB-231 cells. Therefore, further exploration of the interactions between UBL3 and other proteins may facilitate the exploration of the potential effects of UBL3 and diseases.

Alpha-synuclein (α-syn), a highly conserved neuronal protein, is highly enriched in presynaptic nerve terminals. The physiological function of α-syn remains largely unclear; several biochemical activities have been proposed, including regulation of dopamine neuro-transmission and synaptic function/plasticity [9]. α-syn knockout mice do not exhibit a distinct phenotype [10]. Misfolded α-syn is aggregated in α-synucleinopathies, such as Parkinson’s disease (PD), dementia with Lewy bodies [11], and multiple system atrophy [12]. Various factors are involved in the transition of α-syn from a physiological state to pathological aggregation, including genetic mutation [13], protein–protein interactions [14], PTM [15], and oxidative stress [16]. The phosphorylation of α-syn at serine-129 may be important for the formation of inclusions in PD and related α-synucleinopathies [17]. The interactions between α-syn and its protein interactomes play key roles in the pathological accumulation of α-syn. For example, α-syn interacts with synphilin-1 promoting the inclusion formation of α-syn [14]. α-syn interacts with molecular chaperone proteins [18] and protein deglycase DJ-1 [19] preventing the formation of oligomerized α-syn. Exploring potential proteins that can interact with α-syn is an essential direction to elucidate the aggregation mechanism of α-syn and find new therapeutic targets for α-synucleinopathies. The pathological α-syn can be packaged in sEVs for cell-to-cell transport [20]. Until now, it is unknown whether α-syn interacts with UBL3.

Protein–protein interactions play essential roles in most biological processes [21] and these interactions provide a wide spectrum of therapeutic targets for the treatment of human diseases [22]. Many natural products and drugs have been reported to be excellent molecule candidates for stabilizing or inhibiting protein–protein interactions [23]. The interaction between α-syn and the target protein can be affected by different compounds or drugs. For example, the interaction between prolyl oligopeptidase and α-syn was downregulated by KYP-2047, a potent prolyl oligopeptidase inhibitor, reducing the accumulation of α-syn inclusion [24]. Furthermore, the protein–protein interaction of α-syn was downregulated by 03A10, a small molecule from the fruits of *Vernicia fordii* (Euphorbiaceae), inhibiting α-syn aggregation [25]. However, it remains unknown whether the interaction between UBL3 and α-syn will be affected by clinical drugs or chemical compounds.

We hypothesize that UBL3 may interact with α-syn in cells and mediate the loading of α-syn into sEVs. In this study, we aim to investigate whether UBL3 can interact with α-syn in cells, and explore whether the treatments of clinical drugs or chemical compounds affect the interaction between UBL3 and α-syn.

## 2. Materials and Methods

### 2.1. Animals

Age is an important risk factor for neurodegenerative diseases. People usually develop the disease around age 60 or older [26]. Therefore, we choose elder mice with an average age of 22 months to investigate the level of phosphorylated serine-129 (p-S-129). All of the mice in this study were on a C57BL/6J background. *Ubl3* knockout (*Ubl3*^−/−^) mice were acquired from the previously established laboratory colony [3]. C57BL/6J strain wild type (WT) mice (SLC Inc., Hamamatsu, Shizuoka, Japan) were used as a control in this study. Three mice are included in each group. All mice were fed and bred at 12 h of light/dark cycles. The genotypes of mice were confirmed by polymerase chain reaction (PCR) according to our previous report [3].

### 2.2. Antibodies and Drugs

The antibodies used were: anti-p-S-129 α-syn antibody (Abcam, ab59264, 1:1000 dilution), anti-α-syn antibody (BioLegend, 834304, 1:1000 dilution), anti-UBL3 antibody (ABclonal, A4028, 1:1000 dilution), anti-MYC antibody (MBL, M1932, 1:1000 dilution), anti-Flag antibody (MERCK, F7425-.2MG, 1:1000 dilution), horseradish peroxidase (HRP)-conjugated anti-rabbit secondary antibody (Cell signaling, 7074, 1:5000 dilution), biotinylated anti-rabbit immunoglobulin G (Vector Laboratories, BA-1000, 1:1000 dilution). 

Some representative drugs related to neurodegenerative diseases, that are available at our institution, were selected as our screening targets to explore whether they have the potential to influence the interaction of UBL3 with α-syn. Furthermore, some chemical compounds [27,28] and clinical drugs [29] that can affect the formation of α-syn aggregate in vitro were also selected as screening targets to explore whether they affect the interaction. Tyrosine kinase inhibition induces autophagy for neurodegenerative-disease-associated amyloid clearance and epidermal growth factor receptor tyrosine kinase inhibitor (EGFR-TKI) can reduce phosphorylated α-syn pathology [30]. Therefore, certain representative EGFR-TKI drugs that are available at our institution were also selected as our screening targets. All clinically approved drugs, natural products, and other bioactive components, a total of 32, were ordered from the corresponding suppliers (Appendix A) and used for drug screening. All drugs were dissolved in dimethyl sulfoxide (DMSO) (FUJIFILM Wako Pure Chemical Corporation, Osaka, Japan) to 10 mM stock solution.

### 2.3. Immunohistochemistry

Immunohistochemistry was conducted according to a previously published protocol [31]. Briefly, sections were incubated in 3% hydrogen peroxide (FUJIFILM Wako Pure Chemical Corporation, Osaka, Japan) in 1× phosphate-buffered saline (PBS, 0.1 mol/L, pH 7.4) for 15 min after serial deparaffinization, washed with 1× PBS three times, and treated with a solution containing 1% bovine serum albumin (Sigma-Aldrich, St. Louis, MI, USA) in 1× PBS for 1 h at room temperature. Then samples were incubated with primary antibody for 1 h at room temperature. After washing three times with 1× PBS, sections were treated for 1 h with secondary antibody washed three times in 1× PBS, and processed using the avidin–biotin complex (Vector Laboratories, Newark, NJ, USA) and prepared in 1× PBS for 30 min at room temperature. The reaction was visualized using DAB (FUJIFILM Wako Pure Chemical Corporation, Osaka, Japan). Finally, the sections were subsequently counterstained using hematoxylin (FUJIFILM Wako Pure Chemical Corporation, Osaka, Japan), dehydrated in graded alcohols (FUJIFILM Wako Pure Chemical Corporation, Osaka, Japan) (80%, 90%, 100%), transparentized with xylene (FUJIFILM Wako Pure Chemical Corporation, Osaka, Japan), and coverslipped with PathoMount (FUJIFILM Wako Pure Chemical Corporation, Osaka, Japan). Images of immunohistochemistry were acquired using a NanoZoomer 2.0 HT system (Hamamatsu Photonics, Hamamatsu, Shizuoka, Japan). To compare the differences in overall immunoreactive signal intensity between WT and *Ubl3*^−/−^ mice, we randomly intercepted five areas with the same square from the substantia nigra of WT and *Ubl3*^−/−^ mice, respectively. Quantification of the immunoreactivity signal intensity of each intercepted area was determined using the analysis software ImageJ 2.0 (National Institutes of Health, Bethesda, MD, USA).

### 2.4. Plasmids Construction

Following the previous paper [32], for the N-terminal region of *Gaussia princeps* luciferase (Gluc) sequence-tagged UBL3 (NGluc-UBL3) plasmid, the coding sequence of UBL3 (NM_007106) was inserted in the frame after the NGluc sequence in the pCI vector between the XhoI site and the MluI site. For the C-terminal region of Gluc sequence-tagged α-syn (α-syn-CGluc), the coding sequence of α-syn (NM_001146055.2) was inserted in the frame before the CGluc sequence in the pCI vector between the Xho I site and the Mlu I site. For the 3xFlag-UBL3 plasmid, the coding sequence of UBL3 was inserted in the frame after the 3xFlag sequence in the pcDNA3.1-3xFlag vector between the BamHI site and the EcoRI site. For the 6xMYC-α-syn plasmid, we amplified it by in vitro PCR using the corresponding primers. After digestion by XhoI and XbaI, the fragment of α-syn was inserted in the pcDNA3-6xMYC vector after the 6xMYC sequence. For the constructions of “CCVIL” amino acids in C-terminal region-deleted mutant of UBL3 (UBL3∆5), including NGluc-UBL3∆5 and 3xFlag-UBL3∆5, we deleted the last five amino acids (5′-CCVIL-3′) of NGluc-UBL3 and 3xFlag-UBL3 by in vitro site-directed mutagenesis using the corresponding primers (Table 1). The fragments of NGluc-UBL3, NGluc-UBL3∆5, and α-syn-CGluc were tagged with an immunoglobulin kappa secretory signal (IKSS) sequence after the start codon. Sequences of all constructed plasmids were verified by Sanger sequencing. All primers are listed in Table 1.

### 2.5. Cell Culture and cDNA Transfection

Human embryonic kidney (HEK) 293 cells (RIKEN Cell Bank, Tsukuba, Ibaraki, Japan) were cultured in Dulbecco’s modified Eagle’s medium (DMEM, Thermo Fisher Scientific, Waltham, MA, USA) with 10% fetal bovine serum (FBS) (Sigma-Aldrich, St. Louis, MI, USA). Cell cultures were incubated at 37 °C in a 5% CO_2_ humidified incubator. Cells were cultured in culture plates to 80–90% confluence, and transiently transfected with cDNA plasmids using Lipofectamine 2000 transfect reagent (Thermo Fisher Scientific, Waltham, MA, USA) diluted in Opti-MEM reduced serum medium (Thermo Fisher Scientific, Waltham, MA, USA) according to the reagent instructions. 

### 2.6. Cell Treatment

For the detection of UBL3 and α-syn interaction, the cell culture medium (CM) of transfected HEK293 cells was changed to FBS (− Opti-MEM after being transfected with NGluc-UBL3, NGluc-UBL3∆5, and α-syn-CGluc plasmids for 12 h. Then CM and cells were collected after further incubation for 3 days.

For the drug screening, HEK293 cells, after being transfected using NGluc-UBL3 and α-syn-CGluc plasmids for 18 h, were plated into 96-well cell culture plates in 100 μL/well of DMEM (10% FBS) and incubated overnight at 37 °C, in 5% CO_2_ humidified incubator. The next day, all solutions of candidate drugs were diluted in prewarmed DMEM (10% FBS) to 1.5 μM and 15 μM, 1.5-fold of the final concentration. Then 50 μL/well of the old culture medium was replaced with 100 μL of pre-warmed DMEM (10% FBS) containing the different candidate drugs to the final concentrations of 1 μM and 10 μM. For the treatment of MPP^+^, the solution of MPP^+^ (Cayman Chemical, Ann Arbor, MI, USA) was diluted in pre-warmed DMEM (10% FBS) to 1.5-fold of final concentrations and 50 μL/well of the old culture medium was replaced with 100 μL of pre-warmed DMEM (10% FBS) containing MPP^+^ to the final concentrations of 50 μM, 100 μM, 300 μM, 500 μM, and 600 μM, separately. The treatment of equivalent volume DMSO as the drug solution was set as a negative control. All cell culture plates were further incubated at 37 °C for another 48 h. Then the CM was collected for luminescence intensity assay.

### 2.7. Sample Preparation and Luciferase Assay

All CM were centrifuged at 1200 rpm for 5 min to remove the cell debris. The collected HEK293 cells were lysed using cell lysis buffer (1% [*v*/*v*] Triton X100 (Sigma-Aldrich, St. Louis, MI, USA) in 1× PBS) and then centrifuged at 15,000 rpm for 5 min to get supernatant. After adding 17 μg/mL coelenterazine (Cosmo Bio, Kyodo, Japan) diluted by Opti-MEM into CM and cell lysate (CL) supernatant, luminescence intensity was measured using a microplate reader (BioTek, Winooski, VT, USA) immediately. The luminescence intensity of untreated DMEM (10% FBS) was set as a background. For the drug screening, all luminescence intensities of CM were corrected by subtracting the luminescence intensity of the background. The background-corrected data were used to compute a ratio to the luminescence intensity compared to the background-corrected luminescence intensity of DMSO.

### 2.8. 3-(4,5-Dimethylthiazol-2-yl)-2,5-diphenyl-2H-tetrazolium Bromide (MTT) Assay

In live cells, MTT is a pale-yellow substrate that is cleaved by living cells to yield a dark blue formazan product. The quantity of formazan corresponds to the number of living cells. For transfected HEK293 cells, following the collection of CM for the measurement of luminescence intensity, we added fresh pre-warmed DMEM (10% FBS) medium to 100 μL/well, mixed with 10 μL of MTT reagent (Sigma-Aldrich, St. Louis, MI, USA) to each well, and incubated in the incubator at 37 °C for 4 h. After adding 100 μL/well of isopropanol (FUJIFILM Wako Pure Chemical Corporation, Osaka, Japan) with 0.04 N hydrogen chloride (KANTO Chemical Corporation, Tokyo, Japan) to dissolve the formazan, the absorbance (OD) of each well at a wavelength of 450 nm was detected using the microplate reader, cell viability (OD_Intervention group_ – OD_Blank group_)/(OD_Control group_ – OD_Blank group_). The blank group had only medium without cells; the control group had medium and cells without intervention.

### 2.9. Bicinchoninic Acid (BCA) Assay

According to the instructions of the BCA test kit (Thermo Fisher Scientific, Waltham, MA, USA), we add 20 µL of each standard or 2 µL cell lysis sample diluted in 18 µL water replicate into a microplate well. Add 200 µL of the working solution to each well and mix the plate thoroughly on a plate shaker for 30 s. Incubate at 37 °C for 30 min. Then measure the absorbance at 562 nm on a microplate reader after cooling the plate to room temperature. Subtract the average absorbance measurement of the blank standard replicates from the measurements of all other individual standard and cell lysis sample replicates. Then calculate the protein concentration of each sample using the standard curve.

### 2.10. Co-Immunoprecipitation

For the validation of interaction between UBL3 and α-syn, HEK293 cells were transfected with 3xFlag-UBL3, 3xFlag-UBL3∆5, and 6xMYC-α-syn plasmids in various combinations for 18 h. After replacing with new CM, the transfected HEK293 cells were continually incubated for 36 h, then washed and collected with ice-cold 1× PBS, pelleted by centrifugation at 1200 rpm for 5 min at 4 °C. Cell pellets were resuspended and lysed using 1% Triton lysate buffer (50 mM Tris-HCl [pH 7.4] (NAKALAI TESQUE, Kyoto, Japan), 100 mM NaCl (FUJIFILM Wako Pure Chemical Corporation, Osaka, Japan), and 1% [*v*/*v*] Triton X-100) for 30 min on ice. Cell debris and unbroken cells were removed by centrifugation at 15,000 rpm for 15 min at 4 °C. Protein content of the supernatant was measured using the BCA assay according to the manufacturer’s instructions. The supernatants containing 500 μg of total protein were incubated with 50 μL of anti-Flag tag antibody magnetic beads (FUJIFILM Wako Pure Chemical Corporation, Osaka, Japan) with rotation for 10 h at 4 °C. An amount of 50 μL of 2-mercaptoethanol (−2× sodium dodecyl sulfate (SDS) sample loading buffer (100 mM Tris-HCl [pH 6.8], 4% SDS (NAKALAI TESQUE, Kyoto, Japan), 20% glycerol (FUJIFILM Wako Pure Chemical Corporation, Osaka, Japan), and 0.01% bromophenol blue (FUJIFILM Wako Pure Chemical Corporation, Osaka, Japan)) was added to the beads after the beads were washed three times using ice-cold wash buffer (50 mM Tris-HCl [pH 7.4], 100 mM NaCl), and boiled at 95 °C for 5 min. CL and precipitated proteins were separated by SDS-PAGE for Western blotting analysis.

### 2.11. Immunoblot

All samples, including CM and CL from transfected HEK293 cells and precipitated proteins, were loaded into 12% SDS-PAGE. Then the proteins were transferred to the polyvinylidene difluoride membrane (Cytiva, Tokyo, Japan). Membranes were blocked with shaking for 1 h at room temperature using 0.5% [*w*/*v*] skim milk (NAKALAI TESQUE, Kyoto, Japan) in Tween-20 (+) (FUJIFILM Wako Pure Chemical Corporation, Osaka, Japan) Tris Buffered Saline (TBS-T; 100 mM Tris-HCl [pH 8.0], 150 mM NaCl, 0.5% [*v*/*v*] Tween-20) and then incubated with shaking overnight at 4 °C using the appropriate primary antibodies. The membranes were incubated with HRP-conjugated anti-rabbit secondary antibody with shaking at room temperature for 1 h after three washes in TBS-T. The immunoreactive proteins were developed using the enhanced chemiluminescence kit (Thermo Fisher Scientific, Waltham, MA, USA) and detected on the FUSION FX imaging system (Vilber Lourmat, Collégien, Seine-et-Marne, France).

### 2.12. Statistical Analysis

Measurement data were analyzed using GraphPad Prism 7.0 (GraphPad Software, LaJolla, CA, USA) statistical software, and expressed as mean ± SD (standard deviation). The differences between groups of data were calculated using a Student’s *t* test for unpaired data. Furthermore, the differences between groups of data were calculated using a one-way analysis of variance (ANOVA) and Dunnett’s post hoc test for the comparison of multiple groups. *p* < 0.05 was considered statistically significant. All cell culture experiments were performed in triplicate.

## 3. Results

### 3.1. p-S-129 α-syn Was Upregulated in the Substantia Nigra of Ubl3^−/−^ Mice

To explore whether UBL3 affects α-syn, the expression of p-S-129 α-syn was investigated in the brain tissues of the WT and *Ubl3*^−/−^ mice using anti-p-S-129 α-syn antibody by immunohistochemistry. The immunoreactive signal intensity of p-S-129 α-syn was significantly increased in the substantia nigra of *Ubl3*^−/−^ mice compared to WT mice (*p* = 0.0005) (Figure 1 and Appendix A). The expression of p-S-129 α-syn was upregulated in the substantia nigra of *Ubl3*^−/−^ mice, which suggested that UBL3 affects α-syn.

### 3.2. UBL3 Interacted with α-syn in Cells

Further, we examined whether UBL3 interacts with α-syn using a *Gaussia princeps* based split luciferase complementation assay (SLCA), a powerful approach taking advantage of the reconstruction of the N-terminal and C-terminal fragments of Gluc to detect protein–protein interactions in vitro (Figure 2A) [33]. In this study, we constructed the NGluc-UBL3, NGluc-UBL3∆5, and α-syn-CGluc plasmids. All constructions contain an IKSS sequence after the start codon (Figure 2B), which allows successfully expressed fragments and their interacting complexes in cells to be secreted into the CM.

Then we co-expressed SLCA constructs in HEK293 cells in various combinations. Expressions of the SLCA constructs were confirmed by immunoblotting (Figure 2C). We measured the luminescence intensities of CM and CL from transfected HEK293 cells and found strong luminescence intensities from both fractions in the cells expressing NGluc-UBL3 + α-syn-CGluc, and NGluc-UBL3∆5 + α-syn-CGluc. As a control, the CM and CL from Gluc over-expressing HEK293 cells showed markedly higher luminescence intensities compared to double-expression HEK293 cells. On the other hand, the luminescence intensities of CM and CL from HEK293 cells, expressing NGluc-UBL3, NGluc-UBL3∆5, or α-syn-CGluc, were as low as the background level (Figure 2D).

To validate the interaction between UBL3 and α-syn, we constructed and co-expressed 3xFlag-UBL3, 3xFlag-UBL3∆5, and 6xMYC-α-syn in various combinations in HEK293 cells (Figure 2E). The signal of 6xMYC-α-syn was detected from the co-immunoprecipitate of 3xFlag-UBL3 and 3xFlag-UBL3∆5, while the signal of 6xMYC-α-syn in the co-immunoprecipitate of 3xFlag-UBL3∆5 was less than that of 3xFlag-UBL3 (Figure 2F and Appendix A). These results showed that UBL3 interacts with α-syn in HEK293 cells. 

### 3.3. The Interaction between UBL3 and α-syn in Cells Was Upregulated by the 1-Methyl-4-Phenylpyridinium (MPP^+^) Exposure

MPP^+^, a bioactive derivative of 1-methyl-4-phenyl-1, 2, 3, 6-tetrahydropyridine (MPTP), has been reported to induce toxic aggregation of α-syn in cell models [34] and part of mouse models [35]. Furthermore MPTP has been reported to increase the α-syn immunoreactivity in the neurons of non-human primates [36]. To investigate whether the treatment of MPP^+^ affects the interaction between UBL3 and α-syn, we treated HEK293 cells transfected by NGluc-UBL3 and α-syn-CGluc cDNA with 50 μM, 100 μM, 300 μM, 500 μM, and 600 μM of MPP^+^. We collected CM and assayed the luminescence intensities after being treated with different concentrations of MPP^+^ for 48 h and also assessed the cell viability using an MTT assay. Although the luminescence intensities of CM from transfected HEK293 cells treated with different concentrations of MPP^+^ did not show significant differences (Figure 3A), the treatment with MPP^+^ at concentrations between 100 μM and 600 μM significantly inhibited cell viability in a concentration-dependent manner (Figure 3B). Therefore, to exclude the effect of cell activity on the luminescence intensities of CM, we divided the luminescence intensities by the cell viability to calculate the ratio of luminescence relative to the cell number. The treatment of MPP^+^ at concentrations between 300 μM and 600 μM significantly upregulated the interaction between UBL3 and α-syn in a dose-dependent manner (Figure 3C).

### 3.4. Interaction between UBL3 and α-syn in Cells Was Significantly Downregulated by Osimertinib

We used the HEK293 cells, transfected with NGluc-UBL3 and α-syn-CGluc cDNA, as a drug screening model to screen for drugs or compounds that can regulate the interaction between UBL3 and α-syn in cells. We assessed the luminescence intensities of CM from transfected HEK293 cells in the presence of 32 drugs (1 μM and 10 μM) at 48 h in triplicate. All luminescence intensities of CM under drug treatment were normalized to that of vehicle treatment. Under the treatment with a concentration of 1 µM (Figure 4A), one drug, sulfasalazine, upregulated the luminescence intensities of CM by more than 25%. In contrast, one drug, docetaxel, downregulated the luminescence intensities of CM by more than 25%. Under the treatment with a concentration of 10 µM (Figure 4B), three drugs upregulated the luminescence intensities of CM by more than 25%, sulfasalazine, pemetrexed, and gemcitabine. In contrast, four drugs downregulated the luminescence intensities of CM by more than 25%, methylcobalamin, erlotinib, docetaxel, and osimertinib.

To exclude the effect of drug cytotoxicity on the drug screening results, we treated transfected HEK293 cells for 48 h using selected drugs that significantly affected the interaction between UBL3 and α-syn at concentrations of 1 μM and 10 μM, and assessed the cell viability with an MTT assay. As shown in Figure 5A, the cell viability was significantly decreased by the treatment of docetaxel, gemcitabine, osimertinib, and pemetrexed at concentrations of 10 μM. Thus, the ratios of luminescence intensities of CM to cell viability were calculated (Figure 5B). At the concentration of 1 µM, the interaction between UBL3 and α-syn was upregulated by the treatment of gemcitabine (ratio = 1.37, *p* = 0.0008), while it was significantly downregulated by the treatment of erlotinib (ratio = 0.73, *p* < 0.0001). At the concentration of 10 µM, the interaction was significantly upregulated by the treatment of gemcitabine (ratio = 1.53, *p* < 0.0001), while it was significantly downregulated by the treatment of erlotinib (ratio = 0.72, *p* = 0.015) and osimertinib (ratio = 0.55, *p* < 0.0001).

## 4. Discussion

This study first revealed that UBL3 could interact with α-syn in cells and was upregulated in response to the MPP^+^ exposure. Furthermore, the interaction could also be regulated by the treatment of clinical drugs. These results provided the first evidence that UBL3 may be involved in α-synucleinopathies with the possibility of being a potential therapeutic target.

The expression level of p-S-129 α-syn was upregulated in the substantia nigra of *Ubl3*^−/−^ mice. The phosphorylation of α-syn at S-129 is important for the formation of misfold α-syn in synucleinopathies [17]. Therefore, UBL3 may be related to the formation of misfold α-syn. On the other hand, α-syn can be secreted via sEVs in neurons for self-protection when they suffer cellular stress or pathological injury [37,38,39]. UBL3 plays a role in the sorting of proteins to sEVs by acting as a PTM factor [3]. Thus, these results suggested that the deletion of *Ubl3* may upregulate the formation of misfold α-syn and the UBL3 may play a role in the sorting of α-syn to sEVs.

Ageta et al. reported that UBL3 can modify its protein interactomes through disulfide binding depending on the cysteine residues at its C-terminal [3]. It is interesting to note that our results showed that the interaction between UBL3 and α-syn in HEK293 cells is not completely erased after the deleting mutation of the cysteine residues at its C-terminus. UBL3, as a member of the ubiquitin-like protein family, contains a ubiquitin-like domain. Ubiquitin and ubiquitin-like proteins, such as NEDD8, SUMO, FAT10, and ISG15, are covalently attached to lysine residues of their protein interactomes through the C-terminal glycine residues [40]. Taken together, it is possible that UBL3 interacts with α-syn in cells in another manner, rather than only dependent on cysteine residues at its C-terminal. In future studies, it is important to discover the interaction mechanism between UBL3 and α-syn.

Our results showed that the interaction between UBL3 and α-syn in cells was upregulated by the MPP^+^ exposure. MPP^+^, a key environmental risk factor of PD, has been widely used as a common neurotoxin for both in vivo and in vitro experiments [41]. MPP^+^ exposure is known to disturb mitochondrial respiration by inhibiting the mitochondrial complex I, and this process plays a role in initiating mitochondrial dysfunction [42], which can induce and promote α-syn accumulation [43]. PD-like symptoms and aggregation of α-syn were observed in chronic MPP^+^-exposed rodent models [35]. α-syn was involved in the process of induction of mitochondrial dysfunction by MPP^+^ exposure [44]. These results suggested that the upregulation of interaction between UBL3 and α-syn induced by MPP^+^ exposure might be a response to the mitochondrial dysfunction. However, MPP^+^ exposure can also upregulate the expression of α-syn in SH-SY5Y cells [45]. Furthermore, in MPTP-induced non-human primates and partial rodent models it was reported that only α-syn immunoactivity was observed to be upregulated, without significant Lewy body or Lewy neurite formation [36]. Whether the upregulated interaction affects the accumulation of α-syn will need to be investigated in future studies.

The treatment of osimertinib significantly downregulated the interaction between UBL3 and α-syn in cells. Osimertinib, a third-generation EGFR-TKI, is widely used to treat non-small-cell lung cancer [46]. In recent years, it has been suggested that the EGFR signaling pathway and associated genes possibly play an essential role in dopamine neuron cell death [47]. An exogenous neurotrophic supply of EGFR ligands rescues dopaminergic neurons from cell death induced by neurotoxins, 6-hydroxydopamine, or MPTP [48]. An epidemiological study about the polymorphisms of the human EGFR gene found that rs730437 and rs11506105 polymorphisms of EGFR are possible in association with the susceptibility to PD [49]. The treatment of EGFR inhibitors can significantly reduce the p-S-129 α-syn pathology in mouse brain sections by reducing the level of seeding and propagation of pathological α-syn [30]. In our drug screening results, another EGFR inhibitor, erlotinib, also showed significant downregulation of interaction between UBL3 and α-syn. It is convincing from the view of the propagation pathway of α-syn pathology. α-syn can be secreted and transferred cell to cell via sEVs [20]. sEV-associated α-syn can facilitate the propagation of α-syn pathology through cell-to-cell transfer [32]. Furthermore, UBL3 interacts with its target proteins and regulates the sorting of them into sEVs [3]. Therefore, these results indicated that interaction between UBL3 and α-syn may be associated with the inhibition of the α-syn pathology propagation via sEVs by crosstalk with the EGFR pathway.

In addition, we found that the treatment of gemcitabine significantly upregulated the interaction between UBL3 and α-syn in cells. Gemcitabine, a nucleoside analog, has been widely used as an anticancer drug to treat a variety of cancers [50]. The activated gemcitabine triphosphate complex, formed by linking two phosphates, inhibits DNA synthesis by inhibiting ribonucleotide reductase [51]. The treatment of gemcitabine can induce the initiation of mitochondrial dysfunction [52]. This result is consistent with MPP^+^ exposure, suggesting that the upregulation of interaction between UBL3 and α-syn might be a response to the mitochondrial dysfunction.

This study had some limitations. Firstly, the exact mechanisms by which drug treatment affects the interaction between NGluc-UBL3 and α-syn-CGluc, altering protein expression, degradation, or directly influencing the process of interactions, was not investigated. Secondly, whether the aggregation status of α-syn in HEK293 cells overexpressing α-syn affects the interaction between UBL3 and α-syn remains unstudied. Furthermore, due to technical constraints, it was not possible to determine whether the drug substance treatment would affect the activity of the luciferase itself. On the other hand, the impact of candidate drug treatments and UBL3 itself on the aggregation state of α-syn was also not investigated. In the future, we will further explore these limitations according to the methods summarized by the previous report [53]. In addition, the number and selection range of drugs used for drug screening in this study is limited and more drugs will need to be tested in future studies.

## 5. Conclusions

The results in this study showed that UBL3 interacts with α-syn in cells and the interaction between UBL3 and α-syn is upregulated in response to MPP^+^ exposure. Moreover, it was downregulated by the treatment of EGFR inhibitor osimertinib. These findings provide the first evidence identifying UBL3 as an interacting protein of α-syn and UBL3 may be a new therapeutic option for α-synucleinopathies in the future. This study extends the horizon for further etiological and therapeutic studies of α-synucleinopathies.

## Figures and Tables

**Figure 1 biomedicines-11-01685-f001:**
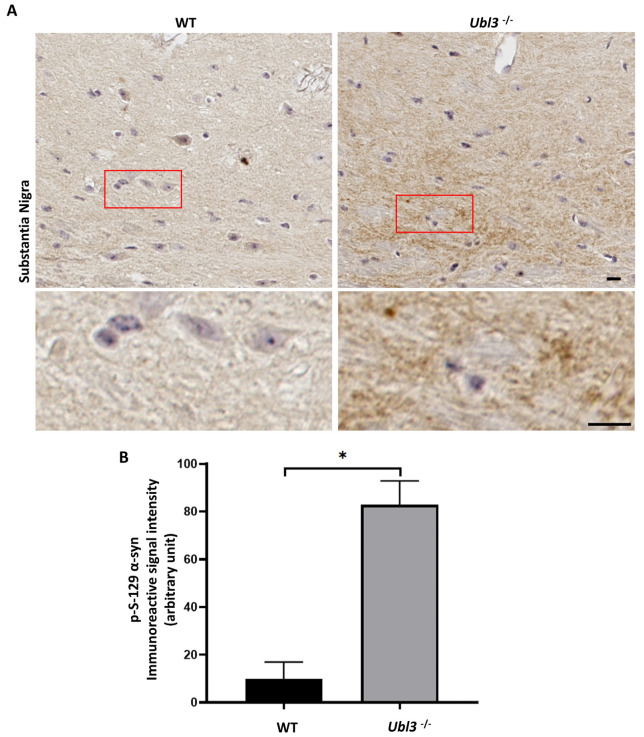
Expression of p-S-129 α-syn was upregulated in the substantia nigra of *Ubl3*^−/−^ mice brain. (**A**) Representative images of immunocytochemistry staining of p-S-129 α-syn in the substantia nigra of WT and *Ubl3*^−/−^ mice. Scale bars: 10 μm. The red boxes represent the selection zones of the magnified pictures below. (**B**) Quantification comparison of the immunohistochemical expression of p-S-129 α-syn in the substantia nigra between WT and *Ubl3*^−/−^ mice. Histograms represent the mean + SD (the number of mice in each group was three). A Student’s t test was performed. *: *p*-value < 0.05; WT: wild type; *Ubl3*^−/−^: *Ubl3* knock out.

**Figure 2 biomedicines-11-01685-f002:**
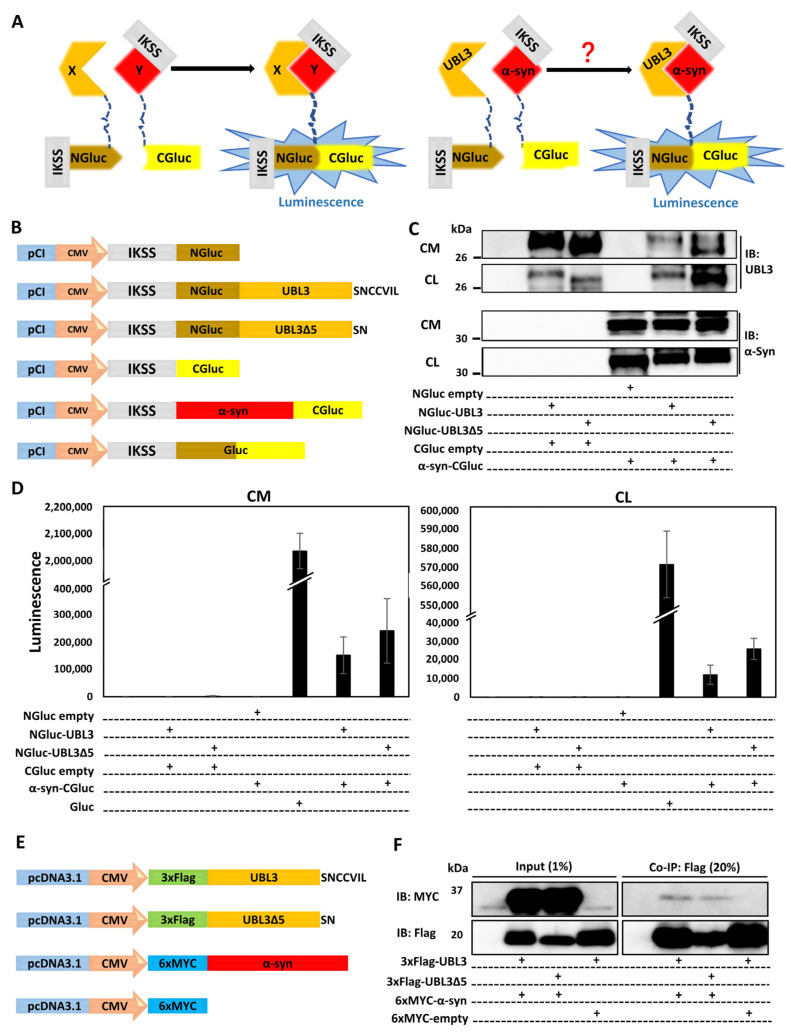
UBL3 interacted with α-syn in cells. (**A**) Schematic representation of the *Gaussia princeps* based split luciferase complementation assay system. Whether there is an interaction between UBL3 and α-syn is still unknown (Red question mark). (**B**) Schematic of split Gluc-tagged proteins, NGluc-UBL3, NGluc-UBL3∆5, and α-syn-CGluc, both containing an IKSS sequence, which allows successfully expressed fragments and their interacting complexes to be secreted into the CM. (**C**) Immunoblotting of CM and CL from transfected HEK293 cells by anti-UBL3 polyclonal antibody and anti-α-syn antibody. (**D**) Luminescence of the CM (left) and CL (right) from transfected HEK293 cells. The luminescence ± SD in triplicate experiments is shown. (**E**) Schematic representation of 3xFlag-UBL3, 3xFlag-UBL3∆5, and 6xMYC-α-syn for co-immunoprecipitation (Co-IP). (**F**) Co-immunoprecipitated 3xFlag-UBL3 and 3xFlag-UBL3∆5 interact with 6xMYC-α-syn. The input lanes are 1% of the sample prior to Co-IP and the Co-IP lanes are 20% of the Co-IP products. IKSS: immunoglobulin kappa secretory signal; CM: cell culture medium; CL: cell lysate.

**Figure 3 biomedicines-11-01685-f003:**
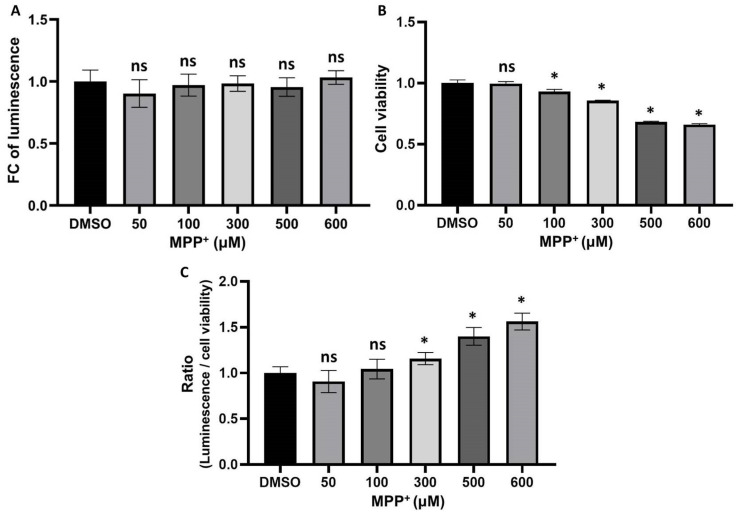
The interaction level in cells was upregulated by the treatment of MPP^+^. (**A**) Luminescence of CM from transfected HEK293 cells, which were treated with 50, 100, 300, 500, and 600 μM of MPP^+^ for 48 h. (**B**) The cell viability of transfected HEK293 cells was treated with different concentrations of MPP^+^ for 48 h. (**C**) The ratios of luminescence divided by cell viability were calculated in triplicate. The luminescence ± SD, cell viability ± SD, and ratio ± SD in triplicate are shown. One-way ANOVA and Dunnett’s post hoc test were performed. ns (non-significant) *p* > 0.05, * *p* < 0.05. FC: fold change; DMSO: dimethyl sulfoxide.

**Figure 4 biomedicines-11-01685-f004:**
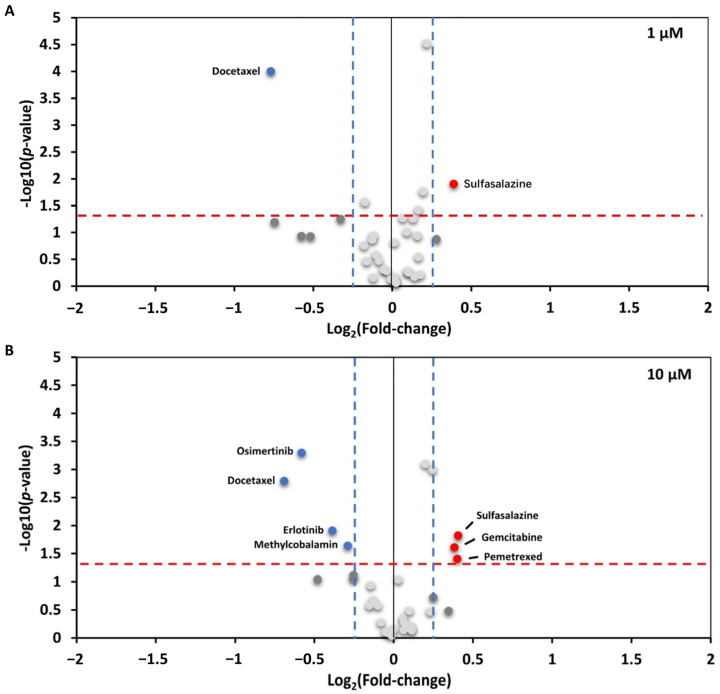
Drug screening results of 32 drugs at concentrations of 1 μM and 10 μM. (**A**) The screen result under a concentration of 1 μM. (**B**) The screen result under a concentration of 10 μM. The volcano plot showed the fold-change (x-axis) versus the significance (y-axis) of 32 drugs. The significance (non-adjusted *p*-value) and the fold-change are converted to −Log10(*p*-value) and Log2(fold-change), respectively. The vertical and horizontal dotted lines show the cut-off of fold-change ± 1.25, and *p*-value = 0.05, respectively. The luminescence of CM from transfected HEK293 cells was upregulated by >1.25-fold with a *p*-value < 0.05 (upper-right, dots colored red) and the luminescence of CM from transfected HEK293 cells was downregulated by < −1.25-fold with *p*-value < 0.05 (upper-left, dots colored blue). The luminescence ± SD in triplicate experiments is shown. One-way ANOVA and Dunnett’s post hoc test were performed. *p* < 0.05 was considered statistically significant.

**Figure 5 biomedicines-11-01685-f005:**
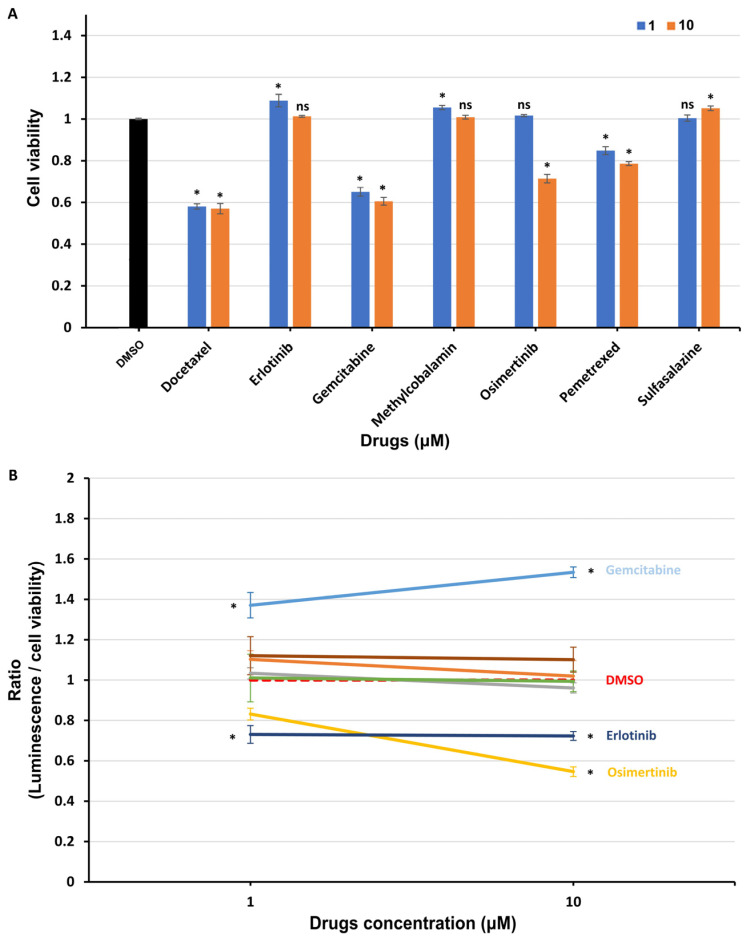
The interaction between UBL3 and α-syn in cells was regulated by the treatment of clinical drugs. (**A**) The cell viability of transfected HEK293 cells treated with selected drugs at concentrations of 1 and 10 μM for 48 h, including docetaxel, erlotinib, gemcitabine, methylcobalamin, osimertinib, pemetrexed, and sulfasalazine. (**B**) The ratio of luminescence divided by cell viability was calculated in triplicate. The cell viability ± SD and the ratio ± SD in triplicate experiments are shown. One-way ANOVA and Dunnett’s post hoc test were performed. ns (non-significant) *p* > 0.05; * *p* < 0.05.

**Table 1 biomedicines-11-01685-t001:** Primer list.

Primer	Sequence
NGluc-UBL3∆5	For 5′-TAAACGCGTGGTACCTCTAGAGTCG-3′
	Rev 5′-ATTACTCTCTCCAGTCTTCTCACGATTCC-3′
3xFlag-UBL3∆5	For 5′-TAAGAATTCTGCAGATATCCATCACAC-3′
	Rev 5′-ATTACTCTCTCCAGTCTTCTCACGATTCC-3′
XhoI-α-syn-XbaI	For 5′-GCACTCGAGGCCACCATGGATG-3′
	Rev 5′-GCCTCTAGATTA GGCTTCAGGTTCGTAGTC-3′

UBL3∆5: CCVIL delete mutant of UBL3, For: Forward, Rev: Reverse.

## Data Availability

All relevant data were reported within the article. Further supporting data will be provided upon a written request addressed to the corresponding author.

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
