# Peer review of "UBL3 Interacts with Alpha-Synuclein in Cells and the Interaction Is Downregulated by the EGFR Pathway Inhibitor Osimertinib"

_biomedicines, 2023, doi:10.3390/biomedicines11061685_

Round 1

Reviewer 1 Report

The quality of the language is fine but can be improved.

Author Response

This manuscript presents a noteworthy discovery by Chen et al. regarding the interaction between alpha-synuclein and Ubiquitin-like 3 (UBL3) in cells, with the potential of being a therapeutic target. While the manuscript does provide valuable insight into this interaction, it would be beneficial to further investigate the functional implications of this interaction, particularly regarding whether it is beneficial or detrimental to cells. Further analysis in this area would enhance the manuscript's contribution to our understanding of the potential therapeutic applications of targeting this interaction.

Response: Thanks very much for taking your time to review this manuscript. We really appreciate all your comments and suggestions! Further investigation regarding the functional implication of the interaction between UBL3 and alpha-synuclein will be explored in our future research. And, we have answered each of your points as follows.

Major Comments:

Point 1: Regarding Figure 1, the results show that the absence of UBL3 leads to an increase in the

phosphorylated levels of alpha-synuclein. However, the selected representative image for the

Ubl3 -/- mice substantia nigra to illustrate this increase displays only two cells with small, stained areas. In contrast, it appears that there are more cells positive for p-S-129 aSyn in the selected field of the WT mice substantia nigra when compared to the Ubl3 -/-. This raises the question of whether the graph in this figure accounts for the number of immunoreactive cells per selected field or the intensity of the signal. As the selected images do not correspond to the bar graph, clarification is needed to reconcile these findings. It would be helpful for the authors to provide more details on the selection criteria for the images and the method used for quantification of the immunoreactive cells, in order to improve the interpretation and reproducibility of the results.

Response 1: In this study, we mainly focused on exploring the effect of Ubl3 deletion on the overall expression level of total p-S-129 aSyn in the substantia nigra of mice, and did not specifically analyze the numbers of immunoreactive positive neurons in substantia nigra of WT and Ubl3 knockout mice. The bar in Figure 1B represented the statistical analysis results of immunoreactive positive signal intensity in the substantia nigra of WT and Ubl3 knockout mice.

According to your helpful comments, we added detailed content about the selection criteria for the images and the method used for quantification of the overall immunoreactive signal intensity: “To compare the differences in overall immunoreactive signal intensity between WT and Ubl3-/- mice, we randomly intercepted five areas with the same square from the substantia nigra of WT and Ubl3-/- mice, respectively. Quantification of the immunoreactivity signal intensity of each intercepted area was determined using the analysis software ImageJ (National Institutes of Health, Bethesda, MD, USA).” (Line 142-145)

We also added more detailed content about the relevant descriptions in the results section 3.1, “The immunoreactive signal intensity of p-S-129 α-syn was significantly increased in the substantia nigra of Ubl3-/- mice compared to WT mice (p = 0.0005) (Figure 1, and Figure S1A, B). “ in Line 270-272.

And, we also modified the name of the Y-axis in Figure 1B to “ p-S-129 α-syn Immunoreactive signal intensity (arbitrary unit)” (Line 276, Figure 1B).

Previous Figure 1B

Revised Figure 1B (Line 276)

Point 2: Figure 2F, the Flag antibody probed membrane image needs to be improved to provide a clearer representation of the results. The band for the IB: Flag Co-IP 3xFlag-UBL3/6xMYC-empty lane appears to be missing, which may be due to the membrane being stripped or improperly developed. This is a critical observation, as the absence of this band raises questions about the specificity of the co-immunoprecipitation results. Therefore, it is essential to provide a clear and unambiguous representation of the results of this experiment, potentially by repeating the experiment, redeveloping the membrane, or using the results of Figure 3 from the original blots file. This would strengthen the validity and reliability of the data presented in this figure.

Response 2: We have replaced the Co-immunoprecipitation results in Figure 2F using the figures in Figure 3 of the original WBs gel results to clearly show all bands (In Figure 2F).

Previous Figure 2F

Revised Figure 2F (Line 307)

Point 3: Regarding the MPP+ model, it is important to note that while this model has been shown to induce dopaminergic neuron degeneration, the surviving neurons demonstrate increased alpha-synuclein (a-Syn) immunoreactivity without forming Lewy bodies or Lewy neurite s, as summarized in the review by Giraldez et al. Therefore, it is less likely that the MPP+ model induces a-Syn aggregation and instead increases the levels of a-Syn protein available for interaction. Given this information, it would be more appropriate for the authors to describe the MPP+ model as a means of upregulating a-Syn levels rather than a proxy for inducing a-Syn aggregates. In light of these findings, it is reasonable to question whether this upregulation of a-Syn would increase its interaction with UBL3. However, if MPP+ did induce a-Syn aggregation, it would be expected that the accessibility of a-Syn C-terminal for interactions would be reduced, potentially decreasing its interaction with UBL3. Therefore, the authors may want to consider addressing this question in their future work to determine the effect of MPP+ on a-Syn aggregation and its subsequent impact on its interaction with UBL3.

Response 3: As you said, it is more appropriate to not just describe the MPP+ exposure-induced a-Syn aggregates in the MPP+ model. Although in the review by Giraldez et al, MPP+ non-human primates and most of the rodent model has been shown to induce dopaminergic neuron degeneration, the surviving neurons demonstrate increased alpha-synuclein (a-Syn) immunoreactivity without forming Lewy bodies or Lewy neurites. We have also observed in other reports that the formation of a-syn aggregation was observed in MPP+ cell models (doi:10.1007/s12035-016-0104-z) and some chronic MPP+ mouse models (doi:10.1016/j.parkreldis.2008.04.008). So we modified the description of MPP+ models in the manuscript to “MPP+, a bio-active derivative of 1-methyl-4-phenyl-1, 2, 3, 6-tetrahydropyridine (MPTP), has been reported to induce toxic aggregation of α-syn in cell models (doi:10.1007/s12035-016-0104-z) and part of mouse models (doi:10.1016/j.parkreldis.2008.04.008). And MPTP has been reported to increase the α-syn immunoreactivity in the neurons of non-human primates (doi:10.1186/s40478-014-0176-9).” at Line 321-324.

And, as you said, the interaction between UBL3 and a-Syn might be upregulated by the upregulation of a-Syn induced by MPP+ exposure. MPP+ exposure has been reported to initiate mitochondrial dysfunction (doi:10.3389/fnagi.2018.00119). And α-syn was essential to the initiation of mitochondrial dysfunction by MPP+ exposure (doi:10.1016/s1470-2045(15)00246-6). Therefore, the upregulation of a-Syn and the upregulated interaction of UBL3 with a-Syn are the response to the MPP+ exposure which was involved in the initiation of mitochondrial dysfunction.

Whether the upregulated interaction of UBL3 with a-Syn induced by MPP+ exposure would be associated with the formation of a-syn aggregation through mitochondrial dysfunction will be explored in future studies.

We also revised the discussion about MPP+ in the discussion section (421-438): “Our results showed that the interaction between UBL3 and α-syn in cells was up-regulated by the MPP+ exposure. MPP+, a key environmental risk factor of PD, has been widely used as a common neurotoxin for both in vivo and in vitro experiments [doi:10.1074/jbc.M005385200]. MPP+ exposure is known to disturb mitochondrial respiration by inhibiting the mitochondrial complex I, and this process plays a role in initiating mitochondrial dysfunction [doi:10.3389/fnagi.2018.00119], which can induce and promote α-syn accumulation [doi:10.1016/j.pneurobio.2018.09.003]. PD-like symptoms and aggregation of α-syn were observed in the chronic MPP+-exposed rodent models [doi:10.1016/j.parkreldis.2008.04.008]. α-syn was involved in the process of induction of mitochondrial dysfunction by MPP+ exposure [doi:10.1016/j.brainres.2009.07.067]. These results suggested that the upregulation of inter-action between UBL3 and α-syn induced by MPP+ exposure might be a response to the mitochondrial dysfunction. However, MPP+ exposure can also upregulate the expression of α-syn in SH-SY5Y cells [doi:10.26355/eurrev_201811_16417]. MPTP-induced non-human primates and partial rodent models were reported that only α-syn immunoactivity was observed to be upregulated, without significant Lewy body or Lewy neurites formation [doi:10.1186/s40478-014-0176-9]. Whether the upregulated interaction affects the accumulation of α-syn. This speculation  will need to be investigated in future studies.”

Minor Comments:

Point 1: In the discussion section, it would be beneficial if the authors could elaborate more on the significance of their Figure 1 results. Specifically, they could further explore why the absence of UBL3 leads to an increase in alpha-synuclein (a-Syn) phosphorylation, and whether this result indicates that the interaction between UBL3 and a-Syn could be beneficial or detrimental to cells. By addressing these points, the authors can provide a more comprehensive understanding of the significance of their results and their potential impact on the field.

Response 1: Thank you for your comment, We have added one more paragraph in the discussion section to discuss Figure 1 results and further explore the potential impact of UBL3 on the level of p-S-129 aSyn.

“The expression level of p-S-129 α-syn was upregulated in the substantia nigra of Ubl3-/- mice. The phosphorylation of α-syn at S-129 is important for the formation of misfold α-syn in synucleinopathies [doi:10.1523/jneurosci.0482-05.2005]. Therefore, UBL3 may be related to the formation of misfold α-syn. On the other hand, α-syn can be secreted via sEVs in neurons for self-protection when they suffer cellular stress or pathological injury [doi:10.1111/j.1471-4159.2010.06695.x., doi:10.1038/s41419-018-0816-2, doi:10.1186/s13024-018-0241-0]. UBL3 plays a role in the sorting of proteins to sEVs by acting as a PTM factor [doi:10.1038/s41467-018-06197-y]. Thus, these results suggested that the deletion of UBL3 may upregulate the formation of misfold α-syn and the UBL3 may play a role in the sorting of α-syn to sEVs.”  (Paragraph 2 in discussion section, Line 402-409)

Reviewer 2 Report

The authors have shown that UBL3 interacts with alpha-syn and such interaction could be upregulated by MPP and downregulated by a number of drugs including TKI such as osimertinib. The methodologies were sound and vigorous and the results supported the conclusion.

Here are a few concerns to be addressed in the revision:

1. Authors should explain the nature (toxic mechanism) of drugs used in Fig.4. Why did the authors pick up these drugs but not others?

2. MPP is both experimental drug and possible environmental risk factor. In this work, it upregulated UBL3 - alpha-syn interaction at very high conc., ie., 0.5 mM . Does MPP conc. get that high in case of intoxication in vivo in humans?

3. For statistics, authors should use ANOVA and post-hoc test for comparisons of multiple groups.

4. In certain parts of the text, authors should use past tense to describe their results.

Mostly fine, but some parts of the results and discussion section, should be in past tense.

Author Response

The authors have shown that UBL3 interacts with alpha-syn and such interaction could be upregulated by MPP and downregulated by a number of drugs including TKI such as osimertinib. The methodologies were sound and vigorous and the results supported the conclusion.

Here are a few concerns to be addressed in the revision:

Response: Thanks very much for taking your time to review this manuscript. We really appreciate all your comments and suggestions! And, we have answered each of your points as follows.

  1. Authors should explain the nature (toxic mechanism) of drugs used in Fig.4. Why did the authors pick up these drugs but not others?

Response 1: We added the reasons for our choice of these drugs in the Methodology section, 2.2. Antibodies and Drugs (Line 114-122): “Sone representative drugs related to neurodegenerative diseases, that are available at our institution, were selected as our screening targets to explore whether they affect -the interaction of UBL3 with α-syn. And, some chemical compounds [doi:10.1007/s00401-007-0332-4, doi:10.1042/bcj20190705] and clinical drugs [doi:10.1111/j.1471-4159.2008.05254.x] can affect the formation of α-syn aggregate in vitro were also select-ed as screening targets to explore ether they affect the interaction. Tyrosine kinase inhibition induces autophagy for neurodegenerative disease-associated amyloid clear-ance, and epidermal growth factor receptor tyrosine kinase inhibitor (EGFR-TKI) can reduce phosphorylated α-syn pathology [doi:10.1007/s13311-021-01017-6]. Therefore, some representative EGFR-TKI drugs available at our institution were also selected as our screening targets.“

The drugs mainly include three groups:

1, Drugs related to the neurodegenerative disease:

A total of 25 clinical drugs associated with neurodegenerative pathologies were found. Among them, drugs available at our institution were selected for the drug screening, including therapeutic drugs for Parkinson’s disease, Alzheimer's disease, and Huntington's disease, and drugs that have been reported to induce Parkinson's symptoms.

2, Drugs affect alpha-synuclein aggregation:

From the Pubmed database, we found 20 chemical compounds and clinical drugs that can affect the formation of α-syn aggregate in vitro. Among them, the drugs available at our institution were selected for drug screening.

3, Anti-cancer drugs:

Three EGFR-TKI drugs available at our institution were selected for drug screening in the study.

However, the method and range of drug selection in this study are limited, and a larger range of more targeted drugs will need to be tested in future studies to explore whether they affect the interaction between UBL3 and alpha-syn. We have also added this limitation in the manuscript- “In addition, the number of drugs used for drug screening in this study is limited and more drugs will need to be tested in future studies.” (Line 477-479)

  1. MPP is both experimental drug and possible environmental risk factor. In this work, it upregulated UBL3 - alpha-syn interaction at very high conc., ie., 0.5 mM . Does MPP conc. get that high in case of intoxication in vivo in humans?

Response 2: Based on the current reports, we are unable to obtain the exact concentration of MPP+ in the blood or other part of the human body in case of intoxication in vivo in humans. MPP+, a toxic metabolite of the closely related compound MPTP, is a widely used neurotoxin to produce a PD model. The treatment of 0-800 μM MPP+ is used to induce aggregation of α-syn in cell culture (DOI:10.1007/s12035-016-0104-z, and DOI: 10.1016/j.febslet.2006.04.057). Therefore, the effects of the treatment with 50, 100, 300, 500, and 600 μM of MPP+ on the interaction between UBL3 and alpha-syn were tested in our experiments.

  1. For statistics, authors should use ANOVA and post-hoc test for comparisons of multiple groups.

Response 3: According to your helpful comment, we have performed a statistical reanalysis of the data in the text using ANOVA and post-hoc test, and showed the relevant results in the corresponding figures (Lines 263-264, 345, 372, and 394). We have also modified the p values to those calculated by ANOVA and post-hoc tests in Line 382, 384, and 386, and in Figure 5.

Previous Figure 5

Modified Figure 5 (Line 388)

  1. In certain parts of the text, authors should use past tense to describe their results.

Response 4: We have revised the tense in some parts of the manuscript using the past tense according to your suggestion.
